# Genomic Assessment of the Contribution of the *Wolbachia* Endosymbiont of *Eurosta solidaginis* to Gall Induction

**DOI:** 10.3390/ijms24119613

**Published:** 2023-06-01

**Authors:** Natalie Fiutek, Matthew B. Couger, Stacy Pirro, Scott W. Roy, José R. de la Torre, Edward F. Connor

**Affiliations:** 1Department of Biology, San Francisco State University, San Francisco, CA 94112, USA; nfiutek7@gmail.com (N.F.); roy@sfsu.edu (S.W.R.); jdelator@sfsu.edu (J.R.d.l.T.); 2Department of Thoracic Surgery, Brigham and Women’s Hospital, Harvard Medical School, Boston, MA 02115, USA; mcouger@bwh.harvard.edu; 3Iridian Genomes Inc., Bethesda, MD 20817, USA; stacy734@yahoo.com

**Keywords:** *Wolbachia*, *Eurosta solidaginis*, gall induction, bacterial symbiosis, phytohormones, effector proteins, metabolic dependence

## Abstract

We explored the genome of the *Wolbachia* strain, *w*Esol, symbiotic with the plant-gall-inducing fly *Eurosta solidaginis* with the goal of determining if *w*Esol contributes to gall induction by its insect host. Gall induction by insects has been hypothesized to involve the secretion of the phytohormones cytokinin and auxin and/or proteinaceous effectors to stimulate cell division and growth in the host plant. We sequenced the metagenome of *E. solidaginis* and *w*Esol and assembled and annotated the genome of *w*Esol. The *w*Esol genome has an assembled length of 1.66 Mbp and contains 1878 protein-coding genes. The *w*Esol genome is replete with proteins encoded by mobile genetic elements and shows evidence of seven different prophages. We also detected evidence of multiple small insertions of *w*Esol genes into the genome of the host insect. Our characterization of the genome of *w*Esol indicates that it is compromised in the synthesis of dimethylallyl pyrophosphate (DMAPP) and S-adenosyl L-methionine (SAM), which are precursors required for the synthesis of cytokinins and methylthiolated cytokinins. *w*Esol is also incapable of synthesizing tryptophan, and its genome contains no enzymes in any of the known pathways for the synthesis of indole-3-acetic acid (IAA) from tryptophan. *w*Esol must steal DMAPP and L-methionine from its host and therefore is unlikely to provide cytokinin and auxin to its insect host for use in gall induction. Furthermore, in spite of its large repertoire of predicted Type IV secreted effector proteins, these effectors are more likely to contribute to the acquisition of nutrients and the manipulation of the host’s cellular environment to contribute to growth and reproduction of *w*Esol than to aid *E. solidaginis* in manipulating its host plant. Combined with earlier work that shows that *w*Esol is absent from the salivary glands of *E. solidaginis*, our results suggest that *w*Esol does not contribute to gall induction by its host.

## 1. Introduction

Bacteria are commonly associated with insect hosts, but the nature of these associations is quite variable. Some bacteria establish mutually symbiotic relationships with their hosts in which, in exchange for shelter and nutrition, the bacteria provide some benefit to the host such as provisioning amino acids or riboflavin or providing immunity against natural enemies [1,2,3]. However, other bacteria may not only use the host for shelter and nutrition but also manipulate the host to enhance bacterial fitness [4,5,6,7,8].

One species of bacterial symbiont that has both mutually symbiotic and manipulative strains is *Wolbachia pipientis*. *Wolbachia* are a group of obligatory intracellular and maternally transmitted bacterial strains which can be found in a range of eukaryotic species, including crustaceans, nematodes, and a high proportion of insect species [5,9,10,11,12]. Most notably, many strains of *Wolbachia* are considered reproductive parasites of their hosts that distort sex ratio and sexual reproduction to favor maternal transmission [4,5,6]. Other strains of *Wolbachia* have evolved to be mutualistic. For example, the *Wolbachia* strain *w*Bm in the nematode *Brugia malayi* has been shown to contribute essential compounds such as heme, nucleotides, and riboflavin to the host [1], while *Wolbachia* strains associated with *Drosophila* confer protection from RNA viruses [3].

A gall is a mass of plant tissue that is formed as a response to a gall inducer, and in which the gall inducer or their offspring resides while feeding on the host plant. A variety of organisms, including insects, mites, fungi, bacteria, protists, and nematodes, are known to induce plant galls [13,14,15,16,17,18,19,20,21]. While the mechanism of gall induction has been established for bacteria, how insects induce plant galls remains an open question.

The fly *Eurosta solidaginis* (Diptera: Tephritidae) induces galls in the apical meristems of *Solidago altissima* L. (Asteraceae) and hosts a strain of *Wolbachia* whose interaction with its host is yet uncharacterized [22]. Herein, we explore the interaction of *Wolbachia* and *E. solidaginis* specifically to determine if the strain of *Wolbachia* associated with *E. solidaginis,* which we name *w*Esol, contributes to gall induction by this species of fly.

The hypothesis proposed by Giron et al. [23] and Bartlett and Connor [24] that symbiotic bacteria might be completely or partially responsible for gall induction by insects was motivated by two observations: several species of bacteria commonly found associated with plant surfaces and in the rhizosphere (*Agrobacterium tumefaciens*, *Allorhizobium vitis*, *Pantoea agglomerans*, *Rhodococcus fascians*, *Pseudomonas savastanoi*, *Streptomyces turgidiscabies*, and others) have strains that themselves induce plant galls [22,25,26]. While none of the gall-inducing strains of these bacterial species have been demonstrated to be symbiotic with insects, strains of some of these genera have been found in association with insects [27,28,29]. Secondly, in a parallel phenomenon known as “green islands”, a *Wolbachia* strain associated with the leaf-mining moth *Phyllonorycter blancardella* (Lepidoptera: Gracillariidae) has been implicated as providing the inducing agent responsible for green island formation [30,31,32,33].

Gall formation by bacteria has been shown to be triggered by the phytohormones indole-3-acetic acid (IAA, an auxin) and cytokinin (CK), which are either secreted by the bacteria into the host plant or genes responsible for their biosynthesis are inserted by the bacteria into the host plant, where they are expressed, leading to localized CK and IAA production [14,15,34,35,36,37]. CK and IAA are the primary hormones involved in cell division and growth in plants, and their synthesis has been demonstrated in gall-inducing bacteria [15,37,38,39].

Cytokinins and auxin (IAA) are widespread in insects [40,41,42,43,44,45,46,47,48,49,50,51,52,53], and they are also found in other gall-inducing organisms [18,20,21,54,55]. Some of the highest concentrations of CKs and IAA have been reported in larvae of *Eurosta solidaginis* [40,41,50].

“Green island” formation by the leaf-mining moth *P. blancardella* has also been attributed to the secretion of CKs by the inducing insect [30,33]. The high concentration of secreted CKs prevents leaf tissues surrounding the feeding site from senescing, thus forming “green islands” in otherwise senescing leaves. *P. blancardella* has been shown to have high whole-body concentrations of CKs and is associated with high concentrations of CKs in the surrounding leaf tissues [30,33]. This moth harbors a strain of *Wolbachia*, which has been implicated as the source of the cytokinins secreted by its host [31,32]. When the adult moths were cured of *Wolbachia*, the offspring were unable to induce “green islands”, suggesting that the *Wolbachia* symbiont is providing the CKs to induce green islands [31].

It is also hypothesized that the secretion of proteinaceous effectors by a gall-inducing insect species might alter gene expression in the plant host, leading to gall induction [56,57,58,59]. However, thus far, secreted effector proteins that have been reported in gall-inducing insects appear to modify the gall structure or color but are not primarily responsible for gall induction [58,59]. Still, some gall-inducing bacteria and fungi do secrete effector proteins that contribute to gall induction [36,60,61].

Hammer et al. [22] examined several gall-inducing and non-gall-inducing insect species using bacterial 16S ribosomal RNA gene amplicon analysis. They conclude that there was no evidence to support the idea that a common bacterial species or a community of bacteria are, in general, associated with gall-inducing insects and absent from non-gall inducing insects. However, they admit that in specific instances, gall induction could potentially be symbiont-dependent.

As an obligate, maternally transmitted symbiont, *Wolbachia* could be stably transmitted and would appear to be a potential candidate bacterium to participate in gall induction in an insect host that is an obligate gall-inducing species. However, no *Wolbachia* have yet been reported to be associated with gall induction, and as an obligate endosymbiont, *Wolbachia* has a much smaller genome with reduced metabolic capacities relative to free-living, gall-inducing bacteria and depends on its host for many metabolic processes [62]. So, while overcoming the hurdle of host-specific transmission, *Wolbachia* may not have the metabolic pathways to provide its insect host with the requisite means to induce a plant gall.

To investigate the potential role of a bacterial symbiont in gall induction, we sequenced the metagenome of *E. solidaginis* and its *Wolbachia* symbiont *w*Esol, and we assembled and annotated the genome of *w*Esol. Annotation of the *w*Esol genome should reveal whether genes in the IAA or CK pathways are present in *w*Esol and potentially identify effector proteins secreted by *w*Esol as well.

## 2. Results

### 2.1. Parsing wEsol and E. solidaginis Contigs

The *E. solidaginis* holobiont metagenome comprised 49,193 contigs and was approximately 1.44 Gbp in length (*Eurosta solidaginis* genome NCBI accession JARSSU01). Using BLASTn and default settings, we detected 1567 metagenomic contigs that had alignments to our database of *Wolbachia* genomes. However, most of these alignments were short (<100 bp) with percent identities less than 70%. We selected a subset of 577 of these contigs with bitscore values ≥ 60, e-values ≤ 1 × 10^−5^, and length ≥ 55 bp for further consideration as candidate *w*Esol contigs. Bootstrap binning of these 577 contigs identified a group of 147 contigs that were provisionally considered *Wolbachia* contigs. One additional contig of 1031 bp that had a BLAST alignment that was 100% identical to a member of our *Wolbachia* database was added to the group of putative *w*Esol contigs. Coverage patterns were consistent with the interpretation that 146 of the initially identified 148 contigs were indeed *w*Esol contigs (*w*Esol had coverage > 6 times higher than *Eurosta*, see below), while the consistent coverage and read mapping of two contigs lead us to classify them as *w*Esol inserts into the host genome. Three additional contigs not identified using BusyBee Web had all CDSs annotated as alphaproteobacterial and showed coverage patterns consistent with *w*Esol contigs so were added to the set of *w*Esol contigs. An additional 58 contigs showed evidence of abrupt coverage changes within contigs, including 4 contigs identified in the initial group by BusyBee Web. Based on mapping of Illumina read pairs and PacBio reads to contigs, we determined that those 4 contigs along with 13 more were chimeras and we added the parts annotated as alphaproteobacterial to our draft genome of *w*Esol (Appendix A).

### 2.2. Features of the wEsol Genome

Tentatively, the genome of *w*Esol (NCBI accession JAQZAU01) consists of 162 contigs with a total length of 1.66 Mbp (Table 1 and Appendix A). The *w*Esol assembly N50 was 19,308 with an L50 of 23. Overall, there was little difference between putative *Wolbachi*a contigs and putative *Eurosta* contigs in GC content or CAI values. GC content for *w*Esol was slightly lower than for the *Eurosta* genome (35.27% versus 37.76%), while CAI values (CAI [mean ± sd]) for *w*Esol [0.612 ± 0.049] were slightly higher than for *Eurosta* [0.572 ± 0.061] or mixed contigs [0.601 ± 0.036]). Overall, *Eurosta* coverage (12–20×) appeared to be substantially lower than *w*Esol coverage (128× Illumina and 189× PacBio), and this was consistent across all three Illumina libraries. Therefore, contig coverage appears to be a reliable means to sort between *Eurosta* and *w*Esol contigs.

For all instances where BLAST identified reading frames within these 162 *w*Esol contigs as putatively eukaryotic, we detected regions of significant similarity of the genome of the predicted eukaryote to *Wolbachia* based on aligning the genomes of these eukaryotes to our large database of *Wolbachia* genomes. A total of 49 CDSs, out of 1878, had best BLAST alignments on 11 arthropod species, with the spider *Trichonephila clavata* (Arachnida: Araneidae) accounting for almost half of those alignments. Therefore, even those reading frames with best BLAST alignments identified as eukaryotic “hypothetical proteins” may arise because of contamination of reference eukaryotic host genomes with proteins from an as yet unreported strain of a *Wolbachia* symbiont.

### 2.3. Horizontal Gene Transfer of wEsol DNA into Eurosta Contigs

Examining the annotations of the eukaryotic contigs with some CDSs showing BLAST alignments to *Wolbachia*, we detected 88 contigs with evidence of insertions of *w*Esol DNA into the host *E. solidaginis* genome (Appendix A). We defined a contig that had potentially experienced horizontal transfer as one possessing CDSs that were classified by BLAST alignments as a mixture of both alphaproteobacterial and eukaryotic CDSs, with no evidence of discontinuity in read coverage across the contig, and with Illumina read pairs and PacBio reads that spanned the alphaproteobacterial and eukaryotic CDSs. In most instances, the putative *w*Esol insertion into the host genome consisted of 4 or fewer CDSs (83%), but in one instance the insertion consisted of 19 CDSs. These 88 contigs were on average 59,322 ± 6302 bp in length (mean ± standard error) and possessed on average 2.77 ± 0.35 alphaproteobacterial CDSs that were 1700 ± 248 bp in total length. The longest insertion was 9733 bp. Many other contigs had BLAST alignments to our database of *Wolbachia* genomes and may represent inserts that had degraded to the extent that recognizable CDSs are no longer present.

### 2.4. Phylogenetic Placement of wEsol and Strain Origin

Phylogenetic inference based on the GTDB markers for our set of *Wolbachia* genomes selected HIVb + F + R5 as the best model with high bootstrap support, particularly for Supergroups A and B (Figure 1). *w*Esol was placed solidly in Supergroup A. Interestingly, *w*Esol was placed in a clade with *Wolbachia* strains from several distantly related hosts, *Carposina sasaki (*Lepidoptera: Carposinidae), *Cardiocondyla obscurior (*Hymenoptera: Formicidae), *Drosophila simulans* (Diptera: Drosophilidae), and *Megacopta cribraria* (Hemiptera: Platispidae), rather than with *Wolbachia* strains from hosts more closely related to *E. solidaginis* (*Rhagoletis pomonella* and *R. zephyria,* Diptera: Tephritidae) or with strains from other gall-inducing hosts, *Pediapsis aceri*s and *Biorhiza pallida* (Hymenoptera: Cynipidae). This pattern suggests that *Wolbachia* is not co-radiating with its hosts (Figure 2).

### 2.5. Comparative Genomics

The predicted proteome of *w*Esol included 1878 proteins, of which 74.9% had predicted functions. The genome of *w*Esol is somewhat larger than its closest relatives and contained an abundance of mobile genetic elements including proteins encoded by the remnants of seven different phages that are likely related to WO, P2, Lambda, phi-C31, HK97, T4, and Mu (Table 2).

Pangenome analysis suggested that the *Wolbachia* strains *w*Esol, *w*CauA, and *w*Ha share a core genome of 846 proteins, with approximately equal numbers of proteins shared among each pair of strains and unique to each strain, except for *w*Esol (Figure 3). *w*Esol possesses 556 proteins (30% of the total number of predicted proteins in *w*Esol) that are not shared with *w*CauA or *w*Ha. Approximately 20% of these *w*Esol specific proteins are from putative mobile genetic elements, 15% are ankyrin-repeat-domain-containing proteins, and another 20% comprise multi-species conserved proteins.

Visualization of regions of sequence similarity (Figure 4) indicates that there are broad regions of similarity within the contigs of *w*Esol and chromosomes of the closely related strains *w*CauA and *w*Ha. However, due to the fragmented assembly of *w*Esol, we can reach no conclusions about the level of synteny above the contig level. Furthermore, since we are ignorant of the true orientation of the contigs of *w*Esol relative to each other, we can only make inferences about the orientation of the regions of synteny within a contig as being direct or inverted relative to *w*CauA or *w*Ha, not for whole contigs.

### 2.6. Potential Contribution of wEsol to Gall Induction by E. solidaginis

#### 2.6.1. Phytohormone Biosynthesis Proteins

Examination of the protein annotations indicates that *w*Esol does not possess an adenylate dimethylallyltransferase gene (EC: 2.5.1.27 or EC: 2.5.1.112) so is incapable of de novo synthesis of cytokinins. Yet, *w*Esol, like other *Wolbachia* strains, possess genes encoding both the MiaA (*t*RNA dimethylallyltransferase, EC: 2.5.1.75) and MiaB (*t*RNA-2-methylthio-N6-dimethylallyladenosine synthase, EC: 2.8.4.3) enzymes, suggesting that *w*Esol could synthesize *t*RNA-bound cytokinins and *t*RNA-bound methylthiolated cytokinins. However, *w*Esol appears to be incapable of synthesizing either of the prenyl group donors (DMAPP or HMBDP) from the MEP pathway. *w*Esol is lacking the *dxs* (EC: 2.2.1.7) gene to initiate the MEP pathway, and its *ispG*/*gcpE* gene (EC: 1.17.7.1 and EC: 1.17.7.3) is rendered non-functional due to an insertion leading to a frameshift mutation. Our examination of a broader set of *Wolbachia* strains (Appendix A) indicates that the majority are not predicted to be able to synthesize DMAPP or HMBDP, with only approximately 20% of strains capable of doing so only if supplied 1-deoxy-d-xylulose-5-phosphate by the host (Figure 5). *w*Esol and the other 63 strains of *Wolbachia* we examined lack the *dxs* gene, as do apparently most of the larger Rickettsiales (KEGG). *ispG/gcpE* is also missing from 66% of the 63 strains we examined, and *ispG/gcpE* is responsible for the penultimate step in the MEP pathway, leading to the synthesis of HMBDP, which is converted to DMAPP by *ispH*.

*w*Esol also appears incapable of synthesizing S-adenosyl-L-methionine (SAM), the methyl group donor required for the production of methylthiolated cytokinins, unless provided L-methionine by its host. While the *w*Esol genome encodes the enzyme to convert L-methionine to SAM (*metK*, EC: 2.5.1.6), it lacks several others involved in methionine synthesis and is incapable of synthesizing L-methionine from either homoserine or aspartate. Our examination of other *Wolbachia* strains found that most strains lacked several enzymes involved in methionine synthesis and none were capable of synthesizing L-methionine (Figure 6 and Appendix A).

*w*Esol possesses a homolog of the enzyme 5′-nucleotidase (EC: 3.1.3.5), which converts AMP to adenosine, so could potentially contribute to conversion of CK nucleotides to nucleosides [68].

We found no evidence in the genome of *w*Esol or other *Wolbachia* strains for any enzymes for the synthesis of IAA from tryptophan. Furthermore, *Wolbachia* strains and bacteria in the Rickettsiales, in general, are unable to synthesize tryptophan [69], so *w*Esol would also need to acquire this amino acid from its insect host.

Examination of the multiple BLASTP alignments of the “hypothetical” and “uncharacterized” proteins revealed no hidden alignments to enzymes annotating as part of the IAA, CK, MEP, SAM, or adenine salvage pathways.

#### 2.6.2. Secreted Effector Proteins

PSORTb predicted only seven proteins as being extracellularly secreted. However, PSORTb tends to predict the location of proteins secreted by *w*Esol as being the host cytoplasm [70]. When using a threshold of 50% probability to classify proteins as Type IV secreted proteins, T4SEfinder predicted that 539 of the 1878 proteins in the *w*Esol genome were Type IV secreted proteins. Approximately half of these predicted Type IV secreted proteins had no functional annotation, and of the remaining proteins, half were encoded by mobile genetic elements, VIR complex proteins, or ankyrin-repeat-domain-containing proteins.

## 3. Discussion

### 3.1. General Features of the wEsol Genome

The *w*Esol genome is larger than the average *Wolbachia* genome but falls within the range of previously determined *Wolbachia* genome sizes (0.86 to 3.1 Mbp) [71,72]. Wu et al. [73] report only three prophages and 138 insertion sequences (e.g., transposases, terminases, maturases) from *w*Mel yet describe its genome as being “overrun” with mobile elements. However, *w*Esol appears to have an even greater abundance of proteins encoded by mobile elements, with evidence of seven different phages and over 300 insertion sequences (Table 1). *w*Esol shared 846 proteins (45% of total predicted proteins) with its closest relatives *w*CauA and *w*Ha but had a large number of unique proteins, many of which were encoded by mobile genetic elements or were ankyrin repeat domain proteins.

Our phylogenetic analysis placed *w*Esol centrally within Supergroup A strains of *W. pipientis*. *w*Esol was placed in a clade, which is also supported by the phylogenetic analysis of over 1000 *Wolbachia* strains performed by [74], with strains from distantly related host species (*Drosophila simulans w*Ha, *Carposina sasakii*, *Megacopta cribaria*, and *Cardiocondyla obscurior*). *w*Esol was not grouped with other *Wolbachia* strains associated with species in the Tephritidae. This pattern of related *Wolbachia* strains not occurring on related hosts is common throughout Supergroups A and B and is illustrated by the crossing lines linking hosts to *Wolbachia* strains in the tanglegram (Figure 2). Furthermore, this pattern suggests that *Wolbachia* strains are not strictly co-radiating with their insect hosts, which is similar to conclusions reached in other studies and suggests that *Wolbachia* strains can be readily transmitted horizontally between hosts [74,75,76,77,78].

*w*Esol also does not group with other strains of *Wolbachia* hosted by gall-inducing insects. We interpret this to indicate that there is no clade of *Wolbachi*a strains that possess traits unique to gall-inducing insect hosts. This result is consistent with the interpretation that *Wolbachia* infections of gall-inducing insect hosts are not associated with the acquisition of the ability to induce plant galls, but represent opportunistic colonization.

We detected evidence of multiple instances of horizontal gene transfer from the *w*Esol genome to the genome of *E. solidaginis*, as previously reported for other *Wolbachia* strains and their hosts [79,80,81,82,83,84]. For *w*Esol and *E. solidaginis*, these insertions usually involve a single or very few CDSs, and slightly over half are annotated as being phage-encoded, tetratricopeptide, or ankyrin repeat domain proteins. Of the remainder, approximately half have functional annotations, and half are hypothetical proteins. As obligate intracellular, maternally inherited endosymbionts, *Wolbachia* has had the opportunity to engage in such transfers. With evidence of *w*Esol having seven phages, although some possibly degraded, perhaps phage transfer of flanking regions may account for some of these insertions [85]. We believe that the existence of the transfer events certainly made assembly of the metagenome more difficult, but without further study and experimentation, we cannot explore the ramifications of these apparent insertions.

### 3.2. Does wEsol Contribute to Gall Induction by Eurosta solidaginis?

Gall induction by insects has been hypothesized to result from the production and secretion of the phytohormones IAA and cytokinin by the insect, or from the secretion of proteinaceous effectors that alter gene expression in the host plant. Both mechanisms are then proposed to lead the cell division and growth in the host plant that constitutes the gall. *Eurosta solidaginis* larvae are known to contain high concentrations of CKs and IAA that are localized to the salivary glands [40,41,50,86]. However, in the absence of an annotated genome for *E. solidaginis*, no examination of secreted effector proteins has yet been conducted.

#### 3.2.1. *w*Esol and Phytohormone Synthesis

*w*Esol is capable of synthesizing CKs. We detected both the *miaA* and *miaB* genes in *w*Esol, which are involved in the prenylation of adenine in *t*RNA at position 37 of the anticodon loop and methylthiolation of the prenyl group, resulting in these two types of modified *t*RNA bound adenine molecules. Upon *t*RNA degradation, CKs and methylthiolated CKs are released, respectively. However, we found *w*Esol and many other strains of *Wolbachia* to be compromised in the synthesis of DMAPP, HMBDP, and SAM, the precursors necessary for such *t*RNA modifications (Figure 5 and Figure 6, and Appendix A). As an obligate intracellular symbiont, *Wolbachi*a has a genome that is smaller than free-living bacteria that induce galls in plants. Like other strains of *Wolbachia* and *Rickettsia*, in *w*Esol the metabolic pathways for the synthesis of DMAPP and SAM are not functional, and *w*Esol must steal these compounds, or in the case of SAM, its precursor L-methionine, from its host to complete these metabolic functions [62,87,88,89]. Therefore, we believe it is highly unlikely that *w*Esol contributes to CK production and secretion by its host.

Andreas et al. [50] report that CKs and methylthiolated CKs are widespread and abundant in insects, not just in gall- and green-island-inducing species, and more extensive work by Tokuda et al. [52] reached similar conclusions for both CKs and IAA. Andreas et al. [50] proposed that insects obtain CKs and methylthiolated CKs from their own biosynthesis of these compounds, although they suggested that symbiotic acquisition was possible in specific instances. A bioinformatic analysis of the transcriptomes of 670 species of Hexapoda and Insecta found that transcripts that encode proteins that are homologs of the enzymes involved in *t*RNA modifications at position A37 and eventually lead to the release of CK and methylthiolated CKs after *t*RNA degradation (EC: 2.5.1.75 and EC: 2.8.4.3, respectively) are widespread among insects [68]. An examination of the BmIPT1 gene from *Bombyx mori* (Lepidoptera: Bombycidae) showed that it is a functional *t*RNA-dimethylallyltransferase gene since it could complement a mutant yeast strain lacking *t*RNA-dimethylallytransferase [90]. Furthermore, isopentenyladenosine (a form of cytokinin) was detected by HPLC in the recombinant yeast strain complemented by BmIPT1 but was absent from the mutant yeast [90]. Finally, the MVA pathway for DMAPP biosynthesis and the enzymes for biosynthesis of SAM from dietary methionine (EC: 2.5.1.6) are also widespread in insects [91,92]. Zhang et al. [33] imply that methylthiolated CKs are largely if not exclusively produced by bacteria. However, Schweizer et al. [93] outline that enzymes for both prenylation of adenine at position 37 of *t*RNA and its subsequent methylthiolation are widespread in all domains of life. Reiter et al. [94] show that the gene CDK5RAP1 in animals is a homolog of the *miaB* gene. Andreas et al. [50] found methylthiolated CKs in insects, and they have been detected in bacteria, protists, fungi, plants, and humans and other mammalian species [95]. Mooi et al. [68] report that transcripts that encode proteins that are homologs of the CDK5RAP1 gene are widespread in Hexapoda and Insecta. Hence, *E. solidaginis* is capable of synthesizing CKs and methylthiolated CKs without assistance from its bacterial symbiont, *w*Esol.

*w*Esol appears unable to synthesize IAA. We found no evidence in the genome of *w*Esol or other *Wolbachia* strains for any enzymes for the synthesis of IAA from tryptophan. Furthermore, *Wolbachia* strains and bacteria in the Rickettsiales, in general, are unable to synthesize tryptophan [69], so *w*Esol would also need to acquire this amino acid from its insect host, which, in turn, would acquire tryptophan via consumption of its host plant. While many species of free-living rhizosphere bacteria synthesize IAA, no *Wolbachia* strains have been reported to do so [96].

Recent work suggests that insects themselves can synthesize IAA from tryptophan [47,97,98,99]. Tokuda et al. [52] found IAA to be ubiquitous in a wide variety of terrestrial arthropods including insects. In a search for the enzymes responsible for IAA biosynthesis in *B. mori*, Miyata et al. [99] reported that two enzymes that constitute a functional pathway for the synthesis of IAA from tryptophan were encoded in the *B. mori* genome and not in the genome of any bacterial associate.

#### 3.2.2. Bacterial Effectors and Gall Induction

Hypotheses that extracellularly secreted effector proteins are responsible for gall induction focus on production and secretion of effectors by the gall-inducing organism, not by its microbial symbionts [56,57,58,59], the idea being that these effectors are secreted into the host plant where they affect expression of host-plant genes, presumably genes that are involved in the cell cycle, leading to increased cell division and growth, hence the plant-tissue growth that becomes the gall. Secretion into the host plant most likely involves secretions associated with feeding—salivary secretions—or secretions associated with oviposition—accessory gland fluids [45,86].

While we cannot rule out the possibility that proteinaceous effectors secreted by *w*Esol or small RNAs play some role in gall induction, we think it unlikely [100]. The bacterial symbiont of *E. solidaginis*, *w*Esol, is predicted to secrete over 500 Type IV secretion system proteins. Both effector proteins and small RNAs are secreted into its host cell and are most likely exclusively involved in the interactions between *w*Esol and its insect host. Most strains of *Wolbachia* are hosted by insects and other arthropod species that do not induce galls in their food plants or are not herbivorous. While many of these predicted Type IV effectors have unknown function, many are associated with mobile genetic elements and VIR complex genes involved in the secretion system. A large number of the predicted effectors are ankyrin repeat domain proteins, which are thought to be involved in protein–protein interactions, rather than interacting with gene expression in the insect host [101,102]. On the other hand, we have not sought to determine the range of small RNAs that *w*Esol may secrete into the host cell, so we are unable to evaluate their potential effects. As an obligate intracellular symbiont, all strains of *Wolbachia* have small genomes that are at most one-third the size of those of free-living, gall-inducing bacteria. *Wolbachia* strains are deficient in a variety of metabolic pathways and must obtain metabolites from their insect hosts [1,62,89]. *Wolbachia* effectors most likely are involved in manipulating the cellular environment of the host to suppress host immune response and to gather metabolites required for growth and reproduction by *w*Esol. Furthermore, many strains of *Wolbachia* distort the sex ratio of their host to facilitate maternal transmission, and effector proteins are likely to be involved in this process. However, in several years of rearing thousands of *E. solidaginis* infected with *w*Esol, we found no evidence of sex ratio distortion, although cytoplasmic incompatibility leading to sex ratio distortion has been detected in other species in Tephritidae with *Wolbachia* symbionts [103].

In this matryoshka-doll-like system of effector protein or small RNA inside a bacterial cell, inside an insect host cell, to be directly responsible for gall induction, effectors of any type produced by a bacterial symbiont must traverse not only the cell membrane of the bacteria but also that of its host, as well as be either produced in or transported to salivary or accessory glands in order to be secreted directly into the host plant of its gall-inducing host. Alternatively, secreted bacterial effectors could conceivably have a knock-on effect by altering gene expression in the insect host, leading the host to produce and secrete effector proteins into the host plant, which in turn lead to gall induction. However, in either instance, the bacterial effectors must have an effect that ultimately results in the secretion of effector proteins by the host insect from either the salivary or accessory glands to achieve an effect on the host plant.

For *E. solidaginis,* which induces the gall via larval feeding, histochemical staining did not reveal any bacterial chromosomes in its salivary glands, which we interpret to indicate that *w*Esol is not present in the salivary glands [86]. The absence of *w*Esol from the salivary glands makes it all the less likely that *w*Esol contributes to gall induction. Bing et al. [104] report that a *Wolbachia* strain alters expression of genes encoding salivary proteins in the plant-eating spider mite, *Tetranychus urticae* (Trombidiformes: Tetranychidae). However, Zhao et al. [105] had previously shown using fluorescence in situ hybridization (FISH) that, unlike in *E. solidaginis,* the *Wolbachia* strain associated with *T. urticae* is associated with the gnathosoma (mouthparts) of its host.

Other gall-inducing species of insects also have associations with strains of *Wolbachia*. Among gall-inducing Cynipidae, some species host *Wolbachia* symbionts, and others do not, yet all induce plant galls [106]. In the gall-inducing *Dryocosmus kuriphilus* (Hymenoptera: Cynipidae), some populations are *Wolbachia*-free, while others have a *Wolbachia* symbiont, yet all populations of *D. kuriphilus* induce galls [107]. A *Wolbachia* symbiont was recently detected in the gall-inducing mite *Fragariocoptes setiger* (Eriophyoidea); however, the authors suggest it is unlikely to be involved in gall induction [108].

## 4. Materials and Methods

### 4.1. Genome Assembly

We sequenced and assembled the metagenome of *E. solidaginis* and its *Wolbachia* symbiont *w*Esol using an Illumina 150 bp-paired end DNA library from a single adult male (NCBI accessions SRR5487544, SRR12231018) and a PacBio Sequel library using 5 SMRT cells based on DNA pooled from the ovaries of 12 adult females (NCBI accession SRR23272044). Metagenome assembly was accomplished using FLYE with default settings for PacBio libraries [109]. Two additional Illumina genomic libraries, one from a pool of first instar larvae (NCBI accession SRR23959424) and one from a pool of adult females (NCBI accession SRR23984855), were also obtained and used to parse contigs but not used for genome assembly.

### 4.2. Identification of wEsol Contigs and Putative Instances of HGT

Identification of *w*Esol contigs from among the more common host-insect contigs consisted of a multi-step process. We initially established a database consisting of at least one assembly for every strain of *Wolbachia* in NCBI (as of 6/1/2020) and the 51 additional *Wolbachia* genomes assembled by Pascar and Chandler [66] from material deposited in the NCBI-Sequence Read Archive. We used BLASTn to search for alignments of our genomic contigs against this database. We then used BusyBee Web and bootstrap binning based on penta-nucleotide frequencies to classify contigs that had alignments to our database of *Wolbachia* genomes [110].

We calculated and plotted the average GC content of each of our candidate contigs using a sliding window (not shown). We computed the Codon Adaptation Index (CAI) for all candidate contigs, using a subset of 72 of the putative *w*Esol contigs that had high similarity to members of our *Wolbachia* database and were at least 5000 bp long to serve as a control group [111]. We also chose approximately 200 contigs with no BLAST alignment to our *Wolbachia* database to serve as a positive control for pure-*Eurosta* contigs. NCBI screening of the *Eurosta* genome and our own BLAST alignments of *Eurosta* contigs against NR yielded no evidence of contamination.

To determine if there was evidence of *Wolbachia* DNA integrated into eukaryotic contigs, we examined the remaining annotated contigs that had significant BLAST alignments to *Wolbachia* yet consisted largely of DNA coding sequences (CDSs) with alignments to Eukaryota.

### 4.3. Genome Annotation

We used an in-house annotation pipeline that predicted DNA coding sequences from the assembled contigs using Prodigal [112]. *t*RNA and *r*RNA genes were predicted using *t*RNAscan-SE, ssu-align, and meta-rna [113,114,115]. Functional annotations were inferred by comparing predicted protein sequences against the NCBI NR database using DIAMOND [116] and against Pfam, COG and TIGRFAM databases using HMMscan [117,118,119,120].

Reading frames in contigs classified as alphaproteobacterial whose best BLAST alignment was to a species of eukaryote were checked by aligning the genomes of the BLAST identified eukaryote species to our large database of *Wolbachia* genomes. High similarity of large sections of the eukaryotic genome to *Wolbachia* was taken as evidence that these species of eukaryotes were potential hosts of an undescribed strain of *Wolbachia*.

### 4.4. Coverage Patterns

We examined all candidate contigs for coverage patterns in all three Illumina libraries and the PacBio library, combined with the coordinates of the BLASTn alignments on our *Wolbachi*a database and the coordinates of regions annotated as either Alphaproteobacteria or Eukaryota. Some contigs showed evidence of abrupt changes in coverage within the contig when plots of read coverage were generated for each contig (not shown). For contigs with abrupt coverage changes, we performed BLASTn alignments of the male and female Illumina libraries against these contigs to determine if read pairs spanned these regions of abrupt changes in coverage. We also examined the alignment of our PacBio SMRT cell sequence reads for these contigs using BWA-mem [121,122]. For contigs that had no Illumina read pairs and 2 or fewer PacBio reads spanning these regions, we classified the contig as chimeric and split the contig at the midpoint of the region. The section of contig identified as *Wolbachia* was added to our set of contigs to comprise the *w*Esol genome and the remainder of the contig was classified as part of the *Eurosta* genome. For those contigs where changes in coverage were spanned by read pairs, we classified the contig as either a *w*Esol or a *Eurosta* contig based on the overall coverage pattern. Some contigs had regions that were annotated as alphaproteobacterial and regions annotated as eukaryotic yet showed no abrupt change in coverage between these regions and had Illumina read pairs and PacBio reads that spanned their boundaries. We treated the alphaproteobacterial regions as instances of horizontal gene transfer from *Wolbachia* into the host genome.

### 4.5. Phylogenetic Placement of wEsol and Host Relationships

To place the *w*Esol strain within the species and determine supergroup membership, we constructed a phylogenetic tree for a subset of the strains of *Wolbachia* for which genomes are available. We selected strains to cover the major *Wolbachia* supergroups and to represent the overall structure of the phylogeny of *Wolbachia* (Table 2). We selected 29 strains from Supergroup A (including *w*Esol), 27 from Supergroup B, and 6 strains from other supergroups, including the Supergroup E strain from *Folsomia candida* Berlin, which we used to root the tree (Collembola: Isotomidae [123]).

To infer phylogenetic relationships among *Wolbachia* strains, we used the 120 bacterial marker proteins, all single-copy genes, from the Genome Taxonomy Database (GTDB) [124,125,126] and employed the GTDB tool kit (GTDB-tk) to extract, determine completeness, and align these proteins from our selected set of *Wolbachia* genomes [127]. We used IQ-TREE 1.6.11 to infer a tree using maximum likelihood (ML) based on a concatenated alignment of the 120 marker proteins [63]. The amino acid substitution model was inferred using jModelTest 2 based on the Bayesian Information Criteria (BIC) [64]. Bootstrap support was estimated using the Ultrafast bootstrap option in IQ -TREE with 1000 replicates [128,129]. Visualization of the tree was accomplished using the R package ape [65,130].

We used the phylogenetic information contained in the Open Tree of Life project (OTL) to produce a cladogram for the insect taxa in our sample that host the Supergroup A strains of *Wolbachia* [131,132,133]. We used the R packages ape and rotl to extract the phylogenetic information and produce an ultrametric, Newick tree [65,67]. To compare the phylogeny of the *Wolbachia* strains to their insect hosts, we generated a tanglegram using the function “cophylo” from the R package phytools [134].

### 4.6. Genome Comparisons

We used the two most closely related strains of *Wolbachia*, *w*CauA from *Carposina sasakii* (Lepidoptera: Carposinidae) and *w*Ha from *Drosophila simulans* (Diptera: Drosophilidae), as a basis for examining the core and pangenome of *w*Esol. We used the OrthoMCL algorithm [135] and default settings from the package get_homologues to classify orthologous proteins from *w*Esol and the two closely related *Wolbachia* strains [136].

To compare the synteny of the *w*Esol genome to that of *w*CauA and *w*Ha, we used Easyfig to visualize the similarity in nucleotide composition of these three genomes [137]. The assembly of the *w*Esol genome is fragmented into multiple contigs; however, the assemblies of both *w*CauA and *w*Ha are represented by a single chromosome. We arbitrarily ordered the contigs of *w*Esol to be as similar in order to the *w*CauA genome as possible. BLASTn alignments of both *w*CauA and *w*Ha to *w*Esol were then visualized. Although the visualization depicts regions of direct and inverted matches, because of the fragmented nature of the *w*Esol genome, regions of similarity can only indicate matches, not their orientation as direct or inverted matches.

### 4.7. Potential Contribution of wEsol to Gall Induction by E. solidaginis

#### 4.7.1. Phytohormone Biosynthesis Proteins

We used the functional annotation of the *w*Esol genome to search for evidence that *w*Esol could contribute to phytohormone production by *E. solidaginis*. Our search of the genome of *w*Esol was guided by the comprehensive set of annotations from KEGG (Kyoto Encyclopedia of Genes and Genomes [138,139,140]) of orthologs for all enzymes with EC and KO numbers from cytokinin and IAA biosynthesis pathways, as well as from pathways for the synthesis of substrates necessary for CK biosynthesis and metabolism (CK, IAA, and substrate synthesis enzymes, Appendix A). We used a similar process to examine the annotated genomes of 63 other strains of *Wolbachia* for enzymes in CK and IAA pathways and from pathways for the synthesis of substrates necessary for CK synthesis (*Wolbachia* strains, Appendix A).

Specifically, we examined all enzymes involved in the zeatin biosynthesis pathway for CK biosynthesis and metabolism (KEGG pathway map 00908), the tryptophan metabolism pathway for the synthesis of IAA (KEGG pathway map 00380), the adenine salvage pathway for the inter-conversion of adenosine monophosphate (AMP) and adenine (KEGG pathway map 00230), the MEP pathway for the production of 2-*C*-methyl-D-erythritol 4-phosphate (dimethylallyl pyrophosphate, DMAPP) and 4-hydroxy-3-methyl-2-but-3-enyl-pyrophosphate (HMBDP) (KEGG pathway map 00900, Appendix A), and the synthesis of S-adenosyl-L-methionine (SAM) from homoserine or aspartate (KEGG pathway map 00270, Appendix A). Enzymes in the adenine salvage pathway are capable of converting cytokinin nucleotides into cytokinin nucleosides and freebase cytokinins, which are proposed to be the active forms of CKs [141,142,143,144,145,146]. DMAPP and HBMPP are prenyl group donors responsible for prenylation of adenine at position 37 of the anticodon in *t*RNA that leads to the release of cytokinins after *t*RNA degradation [141,146,147]. SAM is the methyl group donor for the production of methylthiolated CKs [94,95]. Although tryptophan-independent pathways for the biosynthesis of IAA have been proposed for plants [148,149,150], support for such a pathway is equivocal [151]. Furthermore, no tryptophan-independent IAA pathway has been proposed for bacteria. All confirmed pathways for IAA biosynthesis in plants and bacteria that have enzymes identified involve metabolism of tryptophan.

To determine if predicted proteins in the *w*Esol annotation that had the best BLAST alignments that were identified as “hypothetical” or “uncharacterized” proteins had substantial similarity to enzymes in IAA, CK, MEP, SAM, or adenine salvage pathways, we used Diamond BLASTp to align these proteins against NR, saving the top 25 hits. We then examined these results to determine if, among the best BLAST alignments, the annotations indicated any functional similarity to enzymes in these pathways.

#### 4.7.2. Effector Proteins

To determine which members of the *w*Esol proteome might be extracellularly secreted and potentially interact with host proteins, we used both PSORTb 3.0 to predict sub-cellular and extracellular localization of proteins and T4SE finder to predict Type IV secretion system proteins [70,152].

## 5. Conclusions

Here, we present the first draft genome of the *Wolbachia* endosymbiont from *Eurosta solidaginis.* Our findings suggest that *Wolbachia* is not involved in gall induction in this system. Our characterization of the genome of *w*Esol indicates that it is unlikely to augment *E. solidaginis* in producing CKs and IAA to induce plant galls. Furthermore, in spite of its large repertoire of predicted Type IV secreted effector proteins, these effectors are more likely to contribute to the acquisition of nutrients and the manipulation of the host’s cellular environment to contribute to growth and reproduction of *w*Esol than to aid *E. solidaginis* in manipulating its host plant. Combined with the observations that *w*Esol is absent from the salivary glands of *E. solidaginis* [86], that other gall-inducing insects and mites induce galls without a *Wolbachia* symbiont [106,107,108], and that no microbial strain or community has been found to characterize gall-inducing insects [22], we conclude gall induction by *E. solidaginis* is independent of its symbiosis with *w*Esol and that, in general, gall-induction is symbiont independent. However, we echo Hammer et al. [22] in stating that it remains possible that in specific instances, bacterial symbionts could contribute to gall induction by other insect species. More studies annotating the genomes of bacterial symbionts associated with gall-inducing insects, examining the localization of the bacterial symbiont within the insect host, and functional analysis of secreted effectors could provide critical evidence pertinent to understanding the contribution of microbial symbionts to gall induction by insects.

## Figures and Tables

**Figure 1 ijms-24-09613-f001:**
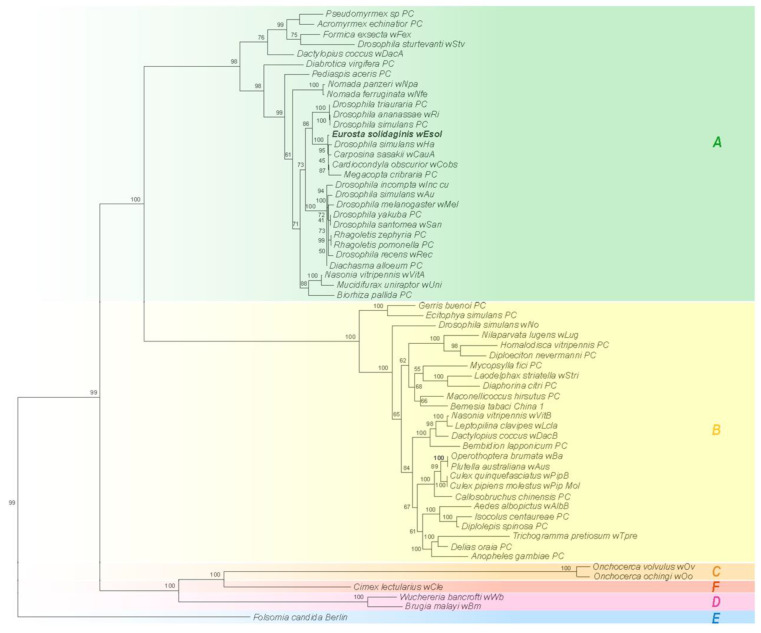
Phylogenetic relationship of 62 *Wolbachia* strains. The tree suggests that *w*Esol is a member of Supergroup A, and since it does not group with other strains from the insect family Tephritidae, that it is not part of a clade co-radiating with other tephritid flies. The tree was constructed using maximum likelihood from a concatenated amino acid alignment of 120 ubiquitous single-copy marker genes (bac120 marker set from GTDB). Numbers on the branches represent the bootstrap support based on 1000 replicates. Tree construction was performed using IQ-TREE 1.6.1 [63]. The final model (HIVb + F + R5) was inferred in jModelTest2 [64] and visualized using ape [65] Supergroups (A–E) are color-coded. The strains assembled by Pascar and Chandler [66] are designated by the suffix PC.

**Figure 2 ijms-24-09613-f002:**
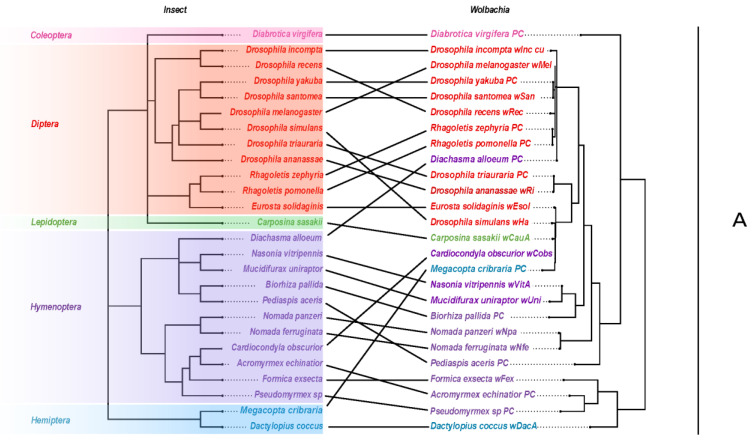
Tanglegram comparing the phylogeny of insect hosts (**left**) to associated *Wolbachia* strains from Supergroup A (**right**). Bold lines indicate links between host and endosymbiont. The insect tree was constructed using the R implementation of the Open Tree of Life project [67]. This tree shows *w*Esol does not group with other *Wolbachia* strains from the Tephritidae family, indicating that *w*Esol was likely horizontally transferred from a non-tephritid host.

**Figure 3 ijms-24-09613-f003:**
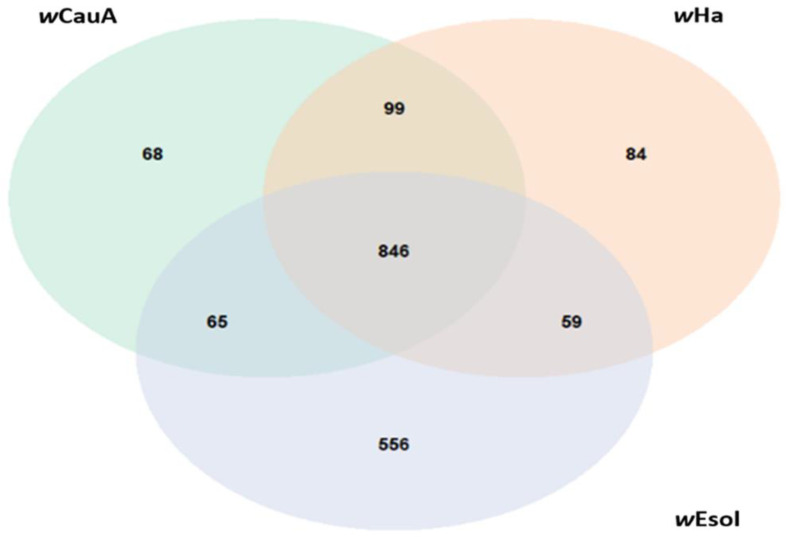
Venn diagram showing the size of the unique and shared components of the proteomes of *w*Esol from *E. solidaginis* and its two closest relatives, *w*CauA from *Carposina sasakii* and *w*Ha from *Drosophila simulans*.

**Figure 4 ijms-24-09613-f004:**
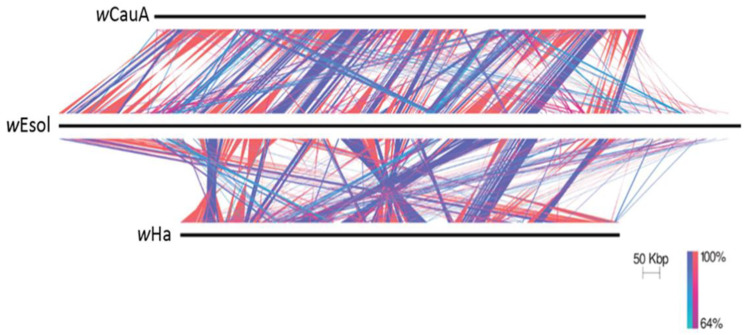
Visualization of regions of nucleotide sequence similarity between *Wolbachia* strains *w*Esol and *w*CauA and *w*Ha. Extensive areas of dark blue and red indicate high synteny within contigs. Because of the fragmented nature of the *w*Esol assembly, no inferences can be made about whether regions of similarity represent direct matches or inverted sequences.

**Figure 5 ijms-24-09613-f005:**
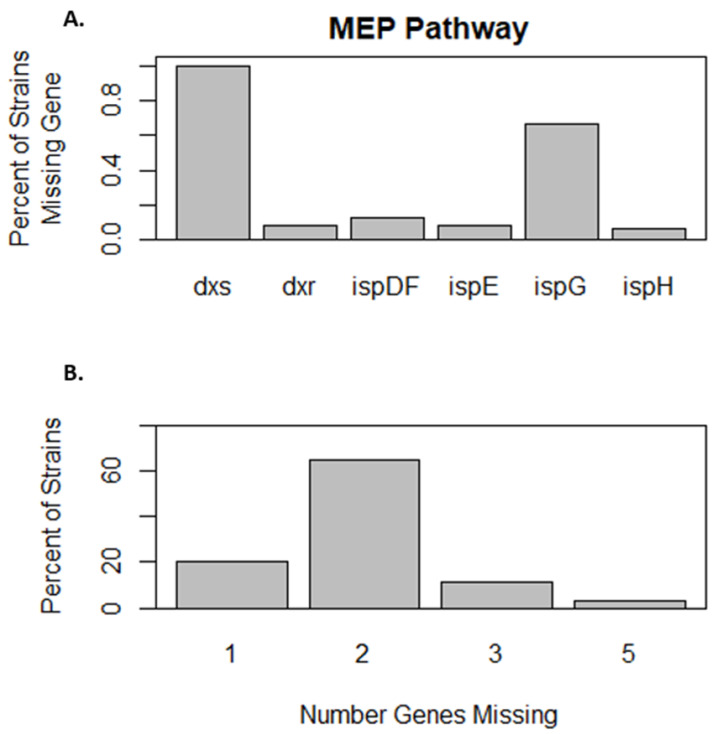
The MEP pathway in *Wolbachia.* (**A**) Enzymes missing from the MEP pathway for DMAPP and HBMPP synthesis in 63 *Wolbachia* strains. (**B**) Extent of enzymatic loss in 63 *Wolbachia* strains.

**Figure 6 ijms-24-09613-f006:**
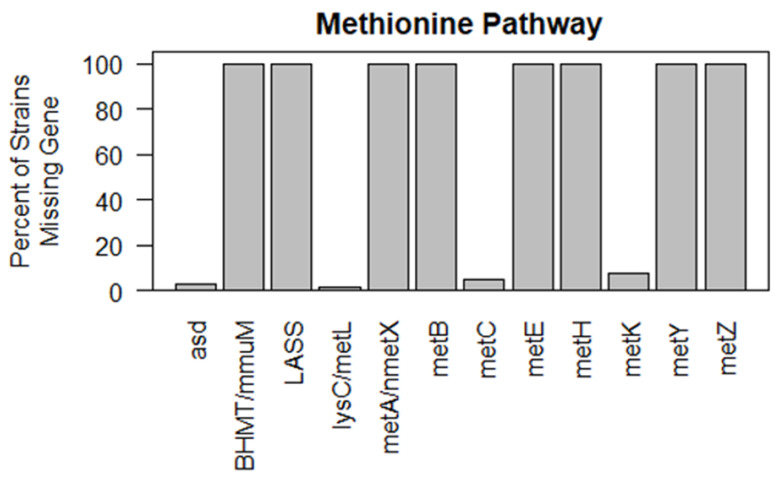
Compromised S-Adenosyl-l-methionine synthesis from aspartate and L-homoserine in 63 *Wolbachia* strains. Most *Wolbachia* strains possess only the *asd*, *LASS*, *metC*, and *metK* genes and are thus compromised in synthesis of S-adenosyl-l-methionine.

**Table 1 ijms-24-09613-t001:** *w*Esol genome features.

Genome size	1,664,961 bp
Predicted CDSs	1878
Average gene length	723 bp
Percent coding	81.5
Assigned function	1406
Multispecies hypothetical	184
Unknown function	472
Hypothetical	245
Transfer RNAs	35
Ribosomal RNAs	1 each of 5S, 16S, and 23S
Prophage	7
GC content	35.3%
Ankyrin-repeat-domain-containing proteins	63
Proteins encoded by mobile genetic elements (transposase, recombinase, integrase)	307

**Table 2 ijms-24-09613-t002:** Accession numbers and supergroups for 62 *Wolbachia* genomes, including *w*Esol, used in the phylogenetic analysis. Host species with “PC” indicate that they were assembled from reads deposited in the GENBANK Sequence Read Archive by Pascar and Chandler [66]. The assemblies generated by [66] are available in their supplemental file.

Species	Strain	Super Group	GenBank Accession
*Pseudomyrmex* sp. PSW-54 PC	-	A	SRR1742977
*Acromyrmex echinatior* PC	-	A	ERR034186, ERR034187
*Formica exsecta*	*w*Fex	A	GCA_003704235.1
*Cardiocondyla obscurior*	*w*Cobs	A	GCA_902713635.1
*Nomada panzeri*	*w*Npa	A	GCA_001675775.1
*Nomada ferruginata*	*w*Nfe	A	GCA_001675785.1
*Pediaspis aceris* 1 PC	-	A	ERR1355090
*Biorhiza pallida* 1 PC	-	A	ERR233308
*Nasonia vitripennis*	*w*VitA	A	GCA_001983615.1
*Mucidifurax uniraptor*	*w*Uni	A	GCA_000174095.1
*Diachasma alloeum* PC	-	A	SRR2042503, SRR2046752
*Drosophila ananassae*	*w*Ri	A	GCA_002907405.1
*Drosophila triauraria* 1 PC	-	A	DRR061000
*Drosophila simulans* 1 PC	-	A	ERR1597896
*Drosophila simulans*	*w*Ha	A	GCA_000376605.1
*Drosophila simulans*	*w*Au	A	GCA_018690095.1
*Eurosta solidaginis*	*w*Esol	A	GCA_029238795.1
*Drosophila melanogaster*	*w*Mel	A	GCA_000008025.1
*Drosophila yakuba* 1 PC	-	A	SRR2318687
*Drosophila santomea*	*w*San	A	GCA_005862095.1
*Drosophila recens*	*w*Rec	A	GCA_000742435.1
*Rhagoletis zephyria* PC	-	A	SRR3670117, SRR3670118, SRR3670120
*Rhagoletis pomonella* PC	-	A	SRR3900841, SRR3901027
*Drosophila incompta*	*w*Inc cu	A	GCA_001758565.1
*Drosophila sturtevanti*	*w*Stv	A	GCA_014107475.1
*Dactylopius coccus*	*w*DacA	A	GCA_001648025.1
*Megacopta cribraria* PC	-	A	SRR1145746
*Diabrotica virgifera virgifera* 1 PC	-	A	SRR1106544, SRR1106897, SRR1106898
*Carposina sasakii*	*w*CauA	A	GCA_006542295.1
*Nasonia vitripennis*	*w*VitB	B	GCA_000204545.1
*Isocolus centaureae* 1 PC	-	B	ERR1359249
*Diplolepis spinosa* 1 PC	-	B	ERR1359308
*Leptopilina clavipes*	*w*Lcla	B	GCA_006334525.1
*Trichogramma pretiosum*	*w*Tpre	B	GCA_001439985.1
*Gerris buenoi* 1 PC	-	B	SRR1197265
*Mycopsylla fici* 1 PC	-	B	SRR2954433
*Nilaparvata lugens*	*w*Lug	B	GCA_007115045.1
*Homalodisca vitripennis* 1 PC	-	B	SRR941995, SRR941996, SRR9419967
*Laodelphax striatellus*	*w*Stri	B	GCA_001637495.1
*Diaphorina citri 1* PC	-	B	SRR183690, SRR189238
*Maconellicoccus hirsutus* PC	-	B	ERR1189167
*Bemesia tabaci* China 1	-	B	GCA_003999585.1
*Dactylopius coccus*	*w*DacB	B	GCA_001648015.1
*Ecitophya simulans* PC	-	B	SRR4301374
*Diploeciton nevermanni* PC	-	B	SRR4342174
*Callosobruchus chinensis* PC	-	B	SRR949786, SRR952345
*Bembidion lapponicum* PC	-	B	SRR2939026
*Operothoptera brumata*	*w*Ba	B	GCA_001266585.1
*Plutella australiana*	*w*Aus	B	GCA_002318985.1
*Delias oraia* PC	-	B	SRR4341246
*Drosophila simulans*	*w*No	B	GCA_000376585.1
*Aedes albopictus*	*w*AlbB	B	GCA_000242415.3
*Culex quinquefasciatus*	*w*PipB	B	GCA_000156735.1
*Culex pipiens molestus*	*wPip Mol*	B	GCA_000208785.1
*Anopheles gambiae* PC	*-*	B	ERR1554834, ERR1554870, ERR1554906
*Onchocerca volvulus*	*w*Ov	C	GCA_000338375.1
*Onchocerca ochingi*	*w*Oo	C	GCA_000306885.1
*Cimex lectularius*	*w*Cle	F	GCA_000829315.1
*Brugia malayi*	*w*Bm	D	GCA_000008385.1
*Wuchereria bancrofti*	*w*Wb	D	GCA_002204235.2
*Folsomia candida Berlin*	-	E	GCA_001931755.2

## Data Availability

All date are available from NCBI or in the Appendix A.

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
