# Peer review of "Genomic Assessment of the Contribution of the Wolbachia Endosymbiont of Eurosta solidaginis to Gall Induction"

_ijms, 2023, doi:10.3390/ijms24119613_

Round 1

Reviewer 1 Report

This is a well-written paper, a pleasure to read (except for the enormous number of citations, which interrupt reading flow - the journal should consider using numbers or superscripts instead). The authors use a genomic approach in an attempt to discern whether Wolbachia bacteria living in an insect contribute to the insect's ability to elicit gall formation on its hostplant. The hypothesis that insect galls on plants are elicited by signals from symbiotic microbes is long-standing. It has been examined a few times.  

It is important to note that the study does not claim to demonstrate that signals from the bacteria (or insect, for that matter) actually function in gall elicitation. The goal was merely to use genomics to determine whether the bacteria present have the genetic and biochemical machinery, as it were, that could provide whatever putative signals are required. What signals are required remains unknown for all gall systems. The authors focus appropriately on the most frequently-suggested possibilities. The study is thus descriptive, but that's important and useful at this stage in trying to understand the galling phenomenon. 

Because of the need to seperate bacterial from insect genes, and the paucity of knowledge about both genomes, the authors needed to combined many approaches and methods to develop their genomic data. This was the most comprehensive, diverse, and detailed study of its type I have ever seen. I have confidence in the data produced and filtered. 

I am somewhat less confident in the authors' interpretation of their results and consideration of alternative hypotheses. 

It seems reasonable that phytohormones might be important signals in gall elicitation, especially since insects and microbes can produce them and some bacteria have been shown to employ them in eliciting gall formation. The authors examined presence of genes encoding enzymes responsible for phytohormone synthesis and found that key elements are missing in the bacteria. They repeatedly raise the alternative that "missing" components - substrates and enzymes - of auxin and cytokinin synthesis could be provided by the host insect. That's common in insect-bacteria symbioses, and would 'complete' the incomplete pathways they found in the bacterial genome. Wolbachia obtains nutrition from the insect host, and that could include substrates needed for hormone synthesis, such as tryptophan. The authors state that Wolbachia "...must steal these compounds, or in the case of SAM its precursor L-methionine, from its host to complete these metabolic functions...Therefore, we believe it is highly unlikely that wEsol contributes to cytokinin production and secretion by its host." But symbiotic bacteria like Wolbachia make their living "stealing" compounds from their hosts. The study found a lack of  "...pathways for the synthesis of substrates necessary for CK biosynthesis". But couldn't the insect provide these substrates? A more explicit discussion this possibility is needed. 

Two important substrates in CK synthesis, DMAPP and HMBDP, are (as the authors point out) also produced via the MVA pathway in insects "...and the enzymes for biosynthesis of SAM from dietary methionine ... are also widespread in insects". This suggests to me that the insect host could supply the necessary substrates. Moreover, these pathways also exist in bacteria, including gall-forming bacteria (https://doi.org/10.3389/fpls.2020.01294). The study doesn't seem to have examined the MVA pathway or alternative ways by which the bacteria could acquire substrates or generate products. I am not suggesting additional lab work here. Finding that signal production is a collaborative effort between bacteria and insect would be interesting. 

But like many readers I am not intimate with these pathways. It's really difficult to follow lists of enzymes, substrates, EC numbers, etc. in the text and understand the conclusions that arise. I strongly urge the authors to provide diagrams of the pathways indicating which steps or substrates cannot be found in the bacteria - and which of these exist in the insect and could be employed by the bacteria. 

The other class of putative signals many feel may be important in gall elicitation is proteins and/or peptides. Such elicitors are common among pathogenic bacteria that infect plants as well as animals. We don't know much about the putative elicitors in this system (or others). A criterion for claiming they have signaling activity is whether their gene sequences indicate that they are secreted. Lacking functional annotation, that's a relatively weak criterion, especially since the bacteria are likely using effectors in their interaction with the insect host. In the absence of functional annotation for the bacterial proteins, I don't see how we can rule out an impact on insect gene expression.

The authors point out that a major portion of the putatively secreted bacterial proteins they found are usually involved in protein-protein interactions. That would seem to open the door to interactions with insect proteins involved in hormone production. The authors argue that the bacterial proteins aren't likely to influence gene expression in the insect host. But transcription factors are proteins that regulate gene expression and could interact with bacterial proteins in both the insect and the plant.

Bacterial peptides would have to get into the plant somehow, meaning they would have to pass through the gut, or salivary glands, or potentially be excreted through the integument. The authors dismiss the possibility of pass-through to the insect's  salivary glands and the plant; but why? They found no bacterial chromosomes in salivary glands, but what about proteins? The possibilities here need more attention. Protein signals from bacteria would have to impact plant gene expression, likely by interacting with transcription factors, to elicit gall formation.

The authors make much of an apparent inability on the part of the bacteria to metabolize tryptophan, which is one basis of auxin synthesis. For some reason the authors dismiss the presence of a tryptophan-independent auxin synthesis pathway. 

Trytophan-independent auxin synthesis exists and is fairly common in bacteria. The references cited actually point this out. Authors should check their data for sequences encoding enzymes in the tryptophan-independent pathway to rule this out. 

The authors state " To determine if there was evidence of Wolbachia DNA integrated into eukaryotic contigs, we examined the remaining annotated contigs that had significant BLAST alignments to Wolbachia, yet consisted largely of DNA coding sequences (CDS) with alignments to Eukaryota." I may be confused but this seems to indicate that some contigs aligning with Wolbachia were found in the eukaryotic DNA. I that is true, do they know what genes may be integrated into the insect's genome?

Overall this study supports the limited set of studies failing to find that microbial symbionts are responsible for insect galls. (The authors missed one: https://doi.org/10.1111/mec.16482) This may become an important generalization. The phylogenetic analysis of Wolbachia strains and their hosts is a real plus as well. The paper's highly detailed and comprehensive analysis makes a valuable contribution to our understanding of gall elicitation. My only wish is for a little more detailed consideration of some of their conclusions, including alternative pathways and ways by which the bacteria may obtain subtrates or enzyme functions. The reader's ability to follow their arguments and conclusions would be aided greatly by providing diagrams indicating which steps in hormone synthesis are blocked and which may have alternative sources.  The overall conclusions may stay the same but they need a little more conceptual justification. 

Author Response

Review 1

Comments and Suggestions for Authors

This is a well-written paper, a pleasure to read (except for the enormous number of citations, which interrupt reading flow - the journal should consider using numbers or superscripts instead). The authors use a genomic approach in an attempt to discern whether Wolbachia bacteria living in an insect contribute to the insect's ability to elicit gall formation on its hostplant. The hypothesis that insect galls on plants are elicited by signals from symbiotic microbes is long-standing. It has been examined a few times.  

We apologize for the large number of citations, but our paper touches on a diverse array of topics that necessitates the copious citations. We also apologize for using a more standard style for citations. The journal actually uses numbers and we have revised our manuscript to follow the journal's style.

It is important to note that the study does not claim to demonstrate that signals from the bacteria (or insect, for that matter) actually function in gall elicitation. The goal was merely to use genomics to determine whether the bacteria present have the genetic and biochemical machinery, as it were, that could provide whatever putative signals are required. What signals are required remains unknown for all gall systems. The authors focus appropriately on the most frequently-suggested possibilities. The study is thus descriptive, but that's important and useful at this stage in trying to understand the galling phenomenon. 

Because of the need to seperate bacterial from insect genes, and the paucity of knowledge about both genomes, the authors needed to combined many approaches and methods to develop their genomic data. This was the most comprehensive, diverse, and detailed study of its type I have ever seen. I have confidence in the data produced and filtered. 

I am somewhat less confident in the authors' interpretation of their results and consideration of alternative hypotheses. 

It seems reasonable that phytohormones might be important signals in gall elicitation, especially since insects and microbes can produce them and some bacteria have been shown to employ them in eliciting gall formation. The authors examined presence of genes encoding enzymes responsible for phytohormone synthesis and found that key elements are missing in the bacteria. They repeatedly raise the alternative that "missing" components - substrates and enzymes - of auxin and cytokinin synthesis could be provided by the host insect. That's common in insect-bacteria symbioses, and would 'complete' the incomplete pathways they found in the bacterial genome. Wolbachia obtains nutrition from the insect host, and that could include substrates needed for hormone synthesis, such as tryptophan. The authors state that Wolbachia "...must steal these compounds, or in the case of SAM its precursor L-methionine, from its host to complete these metabolic functions...Therefore, we believe it is highly unlikely that wEsol contributes to cytokinin production and secretion by its host." But symbiotic bacteria like Wolbachia make their living "stealing" compounds from their hosts. The study found a lack of  "...pathways for the synthesis of substrates necessary for CK biosynthesis". But couldn't the insect provide these substrates? A more explicit discussion this possibility is needed. 

The reviewer is correct. It is possible that Wolbachia obtains the substrates to prenylate tRNA, and to acquire L-methionine and tryptophan from its host. As such Wolbachia could possibly contribute to CK, and methylthiolated CKs in the host. However, the absence of enzymes in tryptophan dependent IAA pathways argues that this is not the case for IAA. We argue below that evidence for the 30 year-old tryptophan-independent pathway is equivocal, and that in any event, no enzymes in such a pathway have been documented, so it was impossible to search for homologs (lines 255-260). On the other hand, we make the case that E. solidaginis and other insects are capable of synthesizing CKs, methylthiolated CKs, and IAA, so that the contribution that Wolbachia might make to the pools of these compounds in the host is most likely minimal (lines 461-496). Furthermore, the evidence we cite from our earlier work shows that CK's and IAA are largely restricted to the salivary gland, and that Wolbachia is absent from the salivary glands of E. solidaginis also argues that Wolbachia is not contributing to the pools of these compounds which would require proteins to transport them to the salivary glands (lines 538-546). We have also moderated our prior claim that wEsol was "incapable" of providing CK and IAA to its host, to say that it is "unlikely" because it is absent from the salivary glands (we have changed line 44 and line 458 from "incapable" to "unlikely.".

Two important substrates in CK synthesis, DMAPP and HMBDP, are (as the authors point out) also produced via the MVA pathway in insects "...and the enzymes for biosynthesis of SAM from dietary methionine ... are also widespread in insects". This suggests to me that the insect host could supply the necessary substrates. Moreover, these pathways also exist in bacteria, including gall-forming bacteria (https://doi.org/10.3389/fpls.2020.01294). The study doesn't seem to have examined the MVA pathway or alternative ways by which the bacteria could acquire substrates or generate products. I am not suggesting additional lab work here. Finding that signal production is a collaborative effort between bacteria and insect would be interesting. 

Yes, Wolbachia could obtain DMAPP,  HMBDP, and L-methionine from its host as we discuss in response to the reviewer's previous comment. However, given the absence of Wolbachia from the salivary glands transport of any CKs produced by Wolbachia to the salivary glands becomes the challenge. Furthermore, the MVA pathway does not occur in Wolbachia, only the partial MEP pathway. 

But like many readers I am not intimate with these pathways. It's really difficult to follow lists of enzymes, substrates, EC numbers, etc. in the text and understand the conclusions that arise. I strongly urge the authors to provide diagrams of the pathways indicating which steps or substrates cannot be found in the bacteria - and which of these exist in the insect and could be employed by the bacteria. 

We have now provided diagram of the MEP and the SAM pathways as Supplemental Figures 1 and 2. We had previously assumed that by providing the KEGG pathway map numbers that interested readers would consult those diagrams.  We indicate in red on each figure those enzymes present in wEsol.

The other class of putative signals many feel may be important in gall elicitation is proteins and/or peptides. Such elicitors are common among pathogenic bacteria that infect plants as well as animals. We don't know much about the putative elicitors in this system (or others). A criterion for claiming they have signaling activity is whether their gene sequences indicate that they are secreted. Lacking functional annotation, that's a relatively weak criterion, especially since the bacteria are likely using effectors in their interaction with the insect host. In the absence of functional annotation for the bacterial proteins, I don't see how we can rule out an impact on insect gene expression.

The reviewer is correct. We cannot completely rule out the possibility that Wolbachia manipulates host gene expression via secreted effector proteins. However, we argue that it unlikely because wEsol is not present in the salivary glands from which any effector that would directly modify gene expression in the host plant must be secreted, and where we believe CK and IAA synthesis occurs (and therefore alteration of host gene expression must also occur).  

The authors point out that a major portion of the putatively secreted bacterial proteins they found are usually involved in protein-protein interactions. That would seem to open the door to interactions with insect proteins involved in hormone production. The authors argue that the bacterial proteins aren't likely to influence gene expression in the insect host. But transcription factors are proteins that regulate gene expression and could interact with bacterial proteins in both the insect and the plant.

But CK and IAA seem to be restricted to the salivary glands where wEsol is not present which makes it less likely that wEsol is manipulating host gene expression in a manner that effects CK or IAA synthesis. Again, we admit it is possible, but think it unlikely.

Bacterial peptides would have to get into the plant somehow, meaning they would have to pass through the gut, or salivary glands, or potentially be excreted through the integument. The authors dismiss the possibility of pass-through to the insect's  salivary glands and the plant; but why? They found no bacterial chromosomes in salivary glands, but what about proteins? The possibilities here need more attention. Protein signals from bacteria would have to impact plant gene expression, likely by interacting with transcription factors, to elicit gall formation.

We have not examined the proteome of the salivary glands, but the absence of wEsol chromosomes from the salivary glands argues that wEsol is also absent from the salivary glands making the interaction between wEsol secreted effectors and host enzymes involved in CK and IAA synthesis improbable. 

The authors make much of an apparent inability on the part of the bacteria to metabolize tryptophan, which is one basis of auxin synthesis. For some reason the authors dismiss the presence of a tryptophan-independent auxin synthesis pathway. 

Trytophan-independent auxin synthesis exists and is fairly common in bacteria. The references cited actually point this out. Authors should check their data for sequences encoding enzymes in the tryptophan-independent pathway to rule this out. 

The existence of a tryptophan-independent auxin pathway is controversial in plants, and no confirmation  of such a pathway has been demonstrated in bacteria. Although this pathway was proposed over 30 years ago, no enzymes have ever been identified, so our bioinformatic approach could not search for these enzymes. However, the reviewer is correct. If such a pathway exists in Wolbachia, then Wolbachia could potentially contribute to the pool of IAA in the insect host. However,  once again, IAA is largely restricted to the salivary glands of E. solidaginis and Wolbachia is absent from the salivary glands. So transport of these compounds to the salivary glands would be required for secretion into the host plant to stimulate gall formation.

We have added text to our methods section (lines 255-259) to provide our rationale for not pursuing the tryptophan-independent pathway.

The authors state " To determine if there was evidence of Wolbachia DNA integrated into eukaryotic contigs, we examined the remaining annotated contigs that had significant BLAST alignments to Wolbachia, yet consisted largely of DNA coding sequences (CDS) with alignments to Eukaryota." I may be confused but this seems to indicate that some contigs aligning with Wolbachia were found in the eukaryotic DNA. I that is true, do they know what genes may be integrated into the insect's genome?

Yes, this is true. It is common for Wolbachia DNA to be found inserted into the host genome. Often these genes are inserted into intronic regions and are non-functional. We will examine these insertions in more detail when we annotate the insect genome in our subsequent work.

Overall this study supports the limited set of studies failing to find that microbial symbionts are responsible for insect galls. (The authors missed one: https://doi.org/10.1111/mec.16482) This may become an important generalization. The phylogenetic analysis of Wolbachia strains and their hosts is a real plus as well. The paper's highly detailed and comprehensive analysis makes a valuable contribution to our understanding of gall elicitation. My only wish is for a little more detailed consideration of some of their conclusions, including alternative pathways and ways by which the bacteria may obtain subtrates or enzyme functions. The reader's ability to follow their arguments and conclusions would be aided greatly by providing diagrams indicating which steps in hormone synthesis are blocked and which may have alternative sources.  The overall conclusions may stay the same but they need a little more conceptual justification. 

The citation the reviewer includes has nothing to do with microbial symbiosis and gall induction. We appreciation the reviewer's concerns about our interpretation of the results. We have endeavored to moderate our interpretations. The reviewer outlines possible ways a bacterial symbiont could contribute to gall induction. We have no direct evidence that wEsol does not synthesize CKs or IAA, nor manipulate host gene expression related to CK or IAA synthesis. However, there is growing evidence that insects can synthesize these compounds. Furthermore,  we do have evidence that CK and IAA pools are not co-located with wEsol. These observations argue strongly that the CK and IAA pools most likely results from synthesis by the insect with little or no contribution from the Wolbachia symbiont.       

Reviewer 2 Report

In this paper, the authors construct the first drat genome of Wolbachia endosymbiont from Eurosta solidaginis. The constructed genome showed that this Wolbachia lacks some of the genes for the biosynthesis of DMAPP and SAM. Based on the genomic analysis, the authors concluded that this Wolbachia does not contribute to the gall-induction. The same conclusion has been shown in previous studies, but the construction of the draft genome and genetic considerations for gall-induction potential has novelty.

Here are some things to consider:

<whole paper>

・Adding lines will help the peer review process run smoothly.

・Figure and table numbers need to be corrected.

・Table 3 and 4 may be duplicated with supplementaly tables. It would be very helpful for the reader to have a schematic representation of the metabolic pathways instead of Tables 3 and 4. 

<Chapter Methods>

・"Sample" and "DNA extraction" sections are required. In particular, information of the sample is required to consider whether Wolbachia living in different Eurosta individuals can be considered to be the same species.

Section "Genome assembly"

・The accession number for "Two additional Illumina genomic libraries" may have been forgotten.

・Contigs with no BLAST hit to the Wolbachia database are considered to be pure-Eurosta contigs. However, I think these contigs could be derived from organisms other than Wolbachia or Eurosta. It will be better if there is evidence that pure-Eurosta contigs are derived from Eurosta.

Section "Genome Annotation"

・I couldn't find a description of the results for the second part. If the result is not shown, I don't think it is necessary to write it in the method.

Section "Coverage patterns"

・This part is very important because demonstrating that the constructed draft genome is free of contamination is of great value to this study. I think it would be better if the criteria for judging "abrupt changes in coverage" were clarified.

Section "Phylogenetic placement of wEsol and host relationships"

・Contrary to the descriptions, it seems that the many of the genomes summarized in Table 1 do not appear to be complete.

<Chapter Results>

Section "Parsing wEsol and E. solidaginis contigs”

・The reason why 147 contigs were identified to be considered Wolbachia contigs should be written. Does this mean they were included in the same group as the Wolbachia database contigs?

・The reason  why "One additional contig of 1031bp" was not included in 577 candidate wEsol contigs may be a point of concern to readers.

・This part is very important but difficult to follow. It would be helpful for the readers to have the flow chart on the supplement. It is also important for a proper understanding of this research.

Section "Comparative genomics"

・"849 proteins" may be "846 ptoteins"

<Chapter Discussion>

Section "Bacterial effectors and gall-induction"

It would be nice to briefly state the rationale for "Wolbachia effectors most likely are involved in manipulating the cellular environment of the host to suppress host immune response and to gather metabolites required for growth and repro- duction by wEsol".

Author Response

Review 2

Here are some things to consider:

<whole paper>

・Adding lines will help the peer review process run smoothly.

We apologize for not adding lines numbers to the manuscript draft. We will submit our revised paper in manuscript form and add line numbers. We will also make revisions on the typeset version we were provided. I tried to add line numbers on the typeset version, but it was not possible. 

・Figure and table numbers need to be corrected.

We had left out a reference to the tanglegram (Figure 2) which we have now inserted. After doing so, the figures are all in the order in which they are cited in text. The Supplemental Tables are also cited in the order they are numbered, except we inadvertently left out Supplemental Table 7 and numbered Supplemental Table 8 as if it were Supplemental Table 7. We have inserted the missing Supplemental Table and numbered them correctly. 

・Table 3 and 4 may be duplicated with supplementaly tables. It would be very helpful for the reader to have a schematic representation of the metabolic pathways instead of Tables 3 and 4. 

 There are no Tables 3 and 4, only the Supplemental Tables 3 and 4.

We have  added Supplemental Figures showing the MEP and SAM pathways (new Figures now numbered Supplemental Figure 1 and 2)

 <Chapter Methods>

・"Sample" and "DNA extraction" sections are required. In particular, information of the sample is required to consider whether Wolbachia living in different Eurosta individuals can be considered to be the same species.

 We indicated in the section titled "Genome Assembly" in the methods that we extracted DNA for the Illumina Library from a single adult male, and for the PacBio SMRT cells from the ovaries of a pool of 12 adult females. Earlier work using bacterial 16S ribosomal RNA gene amplicon analysis characterized multiple individual of Eurosta solidaginis as having a single 16s Wolbachia haplotype  (Hammer et al. 2021). 

Section "Genome assembly"

・The accession number for "Two additional Illumina genomic libraries" may have been forgotten.

Not yet deposited in GENBANK, could do into SRA.

・Contigs with no BLAST hit to the Wolbachia database are considered to be pure-Eurosta contigs. However, I think these contigs could be derived from organisms other than Wolbachia or Eurosta. It will be better if there is evidence that pure-Eurosta contigs are derived from Eurosta.

NCBI screening of the Eurosta genome prior to assigning an accession number reported no evidence of contamination. Also BLAST alignments of Eurosta contigs against NR found no evidence of contamination. We have indicated so on lines 157-159 (lines numbers may change).

Section "Genome Annotation"

・I couldn't find a description of the results for the second part. If the result is not shown, I don't think it is necessary to write it in the method.

 The results for this section are on lines 303-311. There were a total of 49 CDS that BLAST alignments to eukaryotes, mostly the spider Trichonephila clavata. However, all 11 species of eukaryotes had large sections of their genomes align to our Wolbachia database, suggesting that there was un-reported bacterial contamination in these genomes.

Section "Coverage patterns"

・This part is very important because demonstrating that the constructed draft genome is free of contamination is of great value to this study. I think it would be better if the criteria for judging "abrupt changes in coverage" were clarified.

We have attempted to clarify our methods for detecting chimeric contigs. We generated plots of read coverage for each putative contig, and then mapped read to those contigs to determine if either Illumina or PacBio reads spanned this abrupt change in coverage.  These methods are described on lines 175-192.

In the Methods it is difficult to explain "abrupt change in coverage" without presenting information that is given in the Results. However, in the results where it is clear that wEsol contigs tend to have coverage >6 times higher than Eurosta, then it is easier to imagine an "abrupt" change in coverage which could be as high as a 6 fold change.

 Section "Phylogenetic placement of wEsol and host relationships"

・Contrary to the descriptions, it seems that the many of the genomes summarized in Table 1 do not appear to be complete.

We removed the word "complete" from our test on line 195.

<Chapter Results>

Section "Parsing wEsol and E. solidaginis contigs”

・The reason why 147 contigs were identified to be considered Wolbachia contigs should be written. Does this mean they were included in the same group as the Wolbachia database contigs?

The new Supplemental Table 7 gives the reasons for including each contig in the wEsol genome. 147 contigs were identified as Wolbachia using Busy Bee web penta-nucleotide bootstrap binning and  1 contig because it had 100%  identity to our database of Wolbachia genomes. Four of the Busy-Bee identified contigs we determined, based on coverage patterns and read mapping that they were chimeric, and we split them as detailed in the Coverage Patterns section of the Methods and in Supplemental Table 7. Three contigs that were not identified by Busy-Bee were included because on their annotations and coverage pattern. Thirteen more contigs not identified by Busy Bee, were identified as chimeric and their Wolbachia portion was identified by examining their annotation, coverage patterns, and read mapping.

・The reason  why "One additional contig of 1031bp" was not included in 577 candidate wEsol contigs may be a point of concern to readers.

As we state its BLAST alignment to our database of Wolbachia genomes was 100%  identical to a member of our Wolbachia database.

・This part is very important but difficult to follow. It would be helpful for the readers to have the flow chart on the supplement. It is also important for a proper understanding of this research.

We have clarified the methods for detecting chimeric contigs, and added Supplemental Table 7 to explain why each contig was included.

 Section "Comparative genomics"

・"849 proteins" may be "846 ptoteins"

We appreciate the reviewer pointing out our error on line 343.

<Chapter Discussion>

Section "Bacterial effectors and gall-induction"

It would be nice to briefly state the rationale for "Wolbachia effectors most likely are involved in manipulating the cellular environment of the host to suppress host immune response and to gather metabolites required for growth and repro- duction by wEsol".

wEsol is absent from the salivary glands where phytohormones are found and are likely synthesized. Therefore, we doubt that secreted effectors from Wolbachia are involved in modifying host gene expression for genes involved in phytohormone production. We also doubt that Wolbachia effectors could directly manipulate host-plant gene expression again because Wolbachia is absent from the salivary glands generating a barrier to their secretion i

Reviewer 3 Report

The manuscript entitled Genomic assessment of the contribution of the Wolbachia endosymbiont of Eurosta solidaginis to gall-induction is a fascinating manuscript that shows that wEsol does not contribute to gall induction by its host. This king of data is quite essential. Overall the manuscript is well-written and presented. 

Figure 1 could be improved is not very visible.

regarding the references make sure that they are according to the journal guidelines. 

Author Response

Review 3

Comments and Suggestions for Authors

The manuscript entitled Genomic assessment of the contribution of the Wolbachia endosymbiont of Eurosta solidaginis to gall-induction is a fascinating manuscript that shows that wEsol does not contribute to gall induction by its host. This king of data is quite essential. Overall the manuscript is well-written and presented. 

Figure 1 could be improved is not very visible.

We have shifted some of the bootstrap support values to make Figure 1 more readable.

regarding the references make sure that they are according to the journal guidelines. 

We have put the citations and references in the journal's format as requested
